# Spatial distribution of advanced stage diagnosis and mortality of breast cancer: Socioeconomic and health service offer inequalities in Brazil

Nayara Priscila Dantas de Oliveira[1], Marianna de Camargo Cancela[2], Luís Felipe Leite Martins[3], Dyego Leandro Bezerra de Souza[4,5]*

1 Postgraduate Programme in Collective Health, Federal University of Rio Grande do Norte–UFRN, Natal, RN, Brazil, 2 Division of Surveillance and Analysis, Coordination of Prevention and Vigilance (CONPREV), Brazilian National Institute Cancer (INCA), Ministry of Health, Rio de Janeiro-RJ, Brazil, 3 Division of Populational Research, Coordination of Prevention and Vigilance (CONPREV), Brazilian National Institute Cancer (INCA), Ministry of Health, Rio de Janeiro, RJ, Brazil, 4 Department of Collective Health, Postgraduate Programme in Collective Health, Federal University of Rio Grande do Norte–UFRN, Natal, RN, Brazil, 5 Faculty of Health Science and Welfare, Research group on Methodology, Methods, Models and Outcomes of Health and Social Sciences (M3O), Centre for Health and Social Care Research (CESS), University of Vic-Central University of Catalonia (UVic-UCC), Vic, Spain

* dysouz@yahoo.com.br

**Data Availability Statement:** All relevant data are within the manuscript and its Supporting Information files.

## Abstract

Breast cancer presents high incidence and mortality rates, being considered an important public health issue. Analyze the spatial distribution pattern of late stage diagnosis and mortality for breast cancer and its correlation with socioeconomic and health service offer-related population indicators. Ecological study, developed with 161 Intermediate Region of Urban Articulation (IRUA). Mortality data were collected from the Mortality Information System (MIS). Tumor staging data were extracted from the Hospital Cancer Registry (HCR). Socioeconomic variables were obtained from the Atlas of Human Development in Brazil; data on medical density and health services were collected from the National Registry of Health Institutions (NRHI) and Supplementary National Health Agency. Global Moran's Index and Local Indicator of Spatial Association (LISA) were utilized to verify the existence of territorial clusters. Multivariate analysis used models with global spatial effects. The proportion of late stage diagnosis of breast cancer was 39.7% (IC 39.4–40.0). The mean mortality rate for breast cancer, adjusted by the standard world population was 10.65 per 100,000 women (± 3.12). The proportion of late stage diagnosis presented positive spatial correlation with Gini's Index (p = 0.001) and negative with the density of gynecologist doctors (p = 0.009). The adjusted mortality rates presented a positive spatial correlation with the Human Development Index (p<0.001) and density of gynecologist doctors (p<0.001). Socioeconomic and health service offer-related inequalities of the Brazilian territory are determinants of the spatial pattern of breast cancer morbimortality in Brazil.

**Funding:** This study was financed by the Coordenação de Aperfeiçoamento de Pessoal de Nível Superior - Brasil (CAPES) - Finance Code 001. The author NAYARA PRISCILA DANTAS DE OLIVEIRA obtained funding in the doctoral course of the Coordenação de Aperfeiçoamento de Pessoal de Nível Superior - Brasil (CAPES). The funders had no role in study design, data collection and analysis, decision to publish, or preparation of the manuscript.

**Competing interests:** The authors have declared that no competing interests exist.

## Introduction

Breast cancer presents the highest incidence and mortality rates in the female population. Estimates indicate increases in the numbers of cases and deaths due to breast cancer, with regional differences related to different political and socioeconomic different contexts in countries and regions [1]. In countries with high development levels, incidence rates for breast cancer in 2020 were 55.8 cases per 100,000 women, being the highest among female malignant neoplasms [2].

In Brazil, the incidence rate of breast cancer is 61.9 cases per 100,000 women, and it is estimated that 59 thousand new cases and almost 29 thousand new deaths will occur until 2025 [2]. Breast cancer presents significant variations in incidence and mortality across Brazilian regions, with geographic differences that follow health-related inequalities of the population [3,4].

When detected early, the malignant breast neoplasms present a good prognosis, with high cure potential. Late stage diagnosis of breast cancer affects the perspectives of survival, being associated with high treatment costs and worse health indicators [5].

Globally, survival trends for breast cancer have increased. In Brazil, despite the high rates of late-stage diagnosis (40.2%) [5], the five-year survival rates for women diagnosed with breast cancer between 2010–2014 was 75.2%. These survival rates were the highest in the last 10 years, but are still lower than the survival rates of other countries, such as Australia (89.5%), United States (90.2%), Argentina (84.4%) and Costa Rica (86.7%) [6].

Health-related inequalities related to the diagnosis and mortality of breast cancer are affected by contextual socioeconomic conditions and the offer and access to health services [7]. In Brazil, there are high social and income-related inequalities [5,8]. The significant territorial extension and accentuated regional socioeconomic diversity contribute to the irregular distribution of health services and technologies in the Brazilian geographic space. There is limited offer and access to healthcare directed to early detection and timely treatment of breast cancer, incapable of meeting the necessities of the population [7].

Although some studies have analyzed the sociodemographic factors that act as catalysts or mitigating agents of inequalities in breast cancer morbimortality [8–11], spatial distribution patterns and associated factors are frequently ignored [7]. The mapping of geographic patterns of late stage diagnosis and mortality of breast cancer in the Brazilian territorial context can help plan, assess, and implement public policies aimed at the control of breast cancer at local and national levels [7,12].

Besides, the analysis of breast cancer focusing on its geographic location and establishing its relationship with external factors, such as socioeconomic conditions and offer of health services to the population, can reveal underexplored results for the late stage diagnosis and mortality of this cancer.

The objective of this study is to analyze the pattern of spatial distribution of late stage diagnosis and mortality of breast cancer and its correlation with socioeconomic population indicators and health service offer in Brazil.

## Methods

### Study design

This is an observational, ecological study that used the 161 Intermediate Regions of Urban Articulation (IRUA) as an analysis unit, defined by the Brazilian Institute of Geography and Statistics (IBGE) in 2013 [13].

The IRUA correspond to an intermediate territorial scale between Federation Units (FU) and the Immediate Geographic Regions of Urban Articulation [13]. IRUA are agglomerates of

neighboring municipalities, which organize the territory from regional capitals or smaller urban centers, taking into consideration the territorial existence of higher complexity urban functions, including health services [13]. This territorial design emphasizes the municipal flows of public and entrepreneurial management, the mobility of population for work and study purposes, and the regions influenced by the cities [13,14].

The choice to employ the IRUA as the territorial delimitation, referencing year 2013, was based on the capacity of depicting the urban articulations and the contextual reality of the evaluated period in the study. The delimitation of the 161 IRUA presents a dynamic character, depicting urban functions established among the Brazilian municipalities [13]. Besides the portrayed contextual reality, the choice of this geographic unit is also related to the quality of data from health information systems. More disaggregated geographic units are challenged by issues related to coverage and under-registry, which compromises the quality of the information generated.

## Study variables and data sources

The outcomes analyzed in this study were the Adjusted Mortality Rates and the proportion of late stage diagnosis of breast cancer, per IRUA, for 2011–2015. Data on malignant breast neoplasm (CID 10—C50) [15] were obtained from the Brazilian Mortality Information System (MIS) [16]. The place of residence was considered, along with age group for the study period. Deaths with no data on residence and age group were excluded.

The number of deaths was corrected, considering redistribution according to sex, age group, completeness of death records, and ill-defined deaths, following Santos & Souza [17]. Crude and adjusted mortality rates (per 100,000) were calculated for the IRUA, according to the standard world population [18,19] using the direct standardization method [20]. The population in the middle of the evaluated period was used as a reference, collected from the population estimates according to the municipality, sex, and age, available from IBGE [21].

The proportion of late stage diagnosis of breast cancer was extracted from the Brazilian Hospital Cancer Registry Integrator (IHCR) [22]. This registry groups standardized data collected by the HCR, which are located in general or specialized cancer hospitals (public, private, or philanthropic) [23]. The IHCR includes 273 hospital information units for the study period [23], with higher coverage in the South region (75.0%) and lower coverage in the Midwest (50.0%) [5].

Cases of malignant breast neoplasms were collected from IHCR for women aged 18–99 years old, diagnosed in 2011–2015. The cases with no data on the TNM staging of the tumor were excluded, along with carcinoma In Situ (TNM 0) cases, and those with no information on the age and residence at the time of diagnosis.

The clinical tumor staging employed the TNM Classification of Malignant Tumors [24], dichotomized in late stage (TNM III and IV) and early stage (TNM I and II). The proportion of late stage diagnosis of breast cancer was calculated for each IRUA.

Socioeconomic population indices, Gini's Index, and the Human Development Index (HDI) were obtained from the Atlas of Human Development in Brazil for 2010, made available by the United Nations Development Programme [25]. These indicators were collected per municipality and then grouped per IRUA, using the weighted average population. Data on medical density and health service offer were extracted from the National Registry of Health Institutions (NRHI) [26] and Supplementary National Health Agency [27], from which specific indicators were calculated for 2013. The denominators of the indicators were extracted from census data, population counts, and population estimates per municipality, sex, and age, carried out by IBGE [21]. Table 1 presents the study variables and corresponding descriptions.

**Table 1. Characteristics and details of the dependent and independent variables for the assessment of the spatial pattern of mortality and late-stage diagnosis of breast cancer in Brazil, 2011–2015.**

| Variable | | | Source | Description |
|---|---|---|---|---|
| Dependent | Mortality | Adjusted rate of breast cancer mortality | SIM Data from 2011 to 2015 | Female mortality rate for breast cancer adjusted by age and standard world population |
| | Late staging | Proportion of breast cancer late-staging | IRHC Data from 2011 to 2015 | Proportion of late-stage diagnosis of breast cancer considering the TNM System for Tumor Classification (TNM III and IV) |
| Independent (Contextual) | Socioeconomic | Gini Index | s | Measures the degree of inequality in the distribution of individuals according to the per capita household income |
| | | Human Development Index HDI | | Statistics constituted by data on life expectancy, education of GDP per capita |
| Independent (Contextual) | Density of professionals and offer of health services | Density of General Practitioners | CNES (January-December 2013) | Ratio between the average number of general practitioners registered by CNES in 2013 and the total population, multiplied by 100,000, per IRUA. |
| | | Density of Gynecologists | | Ratio between the average number of gynecologists registered by CNES in 2013 and the female population, multiplied by 1,000,000, per IRUA. |
| | | Density of Mastologists | | Ratio between the average number of mastologists registered by CNES in 2013 and the female population, multiplied by 1,000,000, per IRUA. |
| | | Density of mammographic equipment | | Ratio between the average number of gynecologists registered by CNES in 2013 and the female population, multiplied by 1,000,000, per IRUA. |
| | | Proportion of private health plan holders | ANS (January-December 2013) | Average of the ratio, expressed in percentage, between the number of private health plan holders and the total population of 2013, per IRUA. |
| | | Basic attention coverage | List of guidelines, goals and indicators 2014 (2013) | Coverage of the Basic Attention Teams in 2013 from the results achieved by the process of establishing the List of Guidelines, Objectives, Goals, and Indicators 2013–2015 of the Ministry of Health, per RIAU. |

## Statistical analysis

The descriptive analysis of data was carried out using geolocation with software TerraView 5.0.0 [28], using the IRUA for the creation of thematic maps. The analysis describes the spatial distribution of the proportions of late stage diagnosis and adjusted mortality rates of breast cancer in the Brazilian territory in 2011–2015.

Global Moran's Index was used to verify the existence of territorial clusters, which is capable of identifying areas with specific spatial dynamics. The Local Indicator of Spatial Association (LISA) was used to identify significant patterns of spatial correlation [29]. In function of the level of significance of LISA, the IRUA were classified as positively correlated, when the region presents neighbors with similar values (High-high, Low-low), or negatively correlated when the values of the neighboring regions are different (High-low, Low-high). Spatial analysis employed first order queen contiguity.

According to the spatial autocorrelation identification, the independent variables that presented a statistically significant correlation with the dependent variables of the study and non-colinear variables (correlation<0.7) were selected to participate in the spatial regression multivariate analysis.

Multivariate analysis used the Spatial Error Model, which indicates global spatial effects. The decision of the final model considered the highest values of the likelihood log, and lowest values for Akaike's Information Criterion and the Schwarz Information Criterion [29]. The final multivariate model included statistically significant variables and those with theoretical plausibility for inclusion in the statistical model.

The residues generated were analyzed by Moran's I and data dispersion histogram to verify the elimination of spatial correlation after the execution of the multivariate statistical model. The statistical models and calculation of Moran's I and LISA employed Software GeoDa version 1.14 [30].

This study was carried out with secondary data obtained with health information systems, publicly available, which prevents the identification of individuals. Therefore the approval of a Research Ethics Committee (REC) was not necessary, according to Resolution 580/2018 [31].

## Results

In Brazil, for the analyzed period, the IHCR registered the diagnosis of 195,201 cases of malignant breast neoplasms in women aged 18–99 years old. The proportion of late stage diagnosis was 39.7% (IC 39.4–40.0), varying across the Brazilian regions. The mean adjusted mortality rate for breast cancer, considering the world population, was 10.65 per 100,000 women with a standard deviation of 3.12.

Fig 1 presents the spatial distribution of the proportion of late stage diagnosis and the adjusted mortality rates of breast cancer for the 161 IRUA, for 2011–2015.

The existence of spatial autocorrelation between the proportion of late stage diagnosis and adjusted mortality rates of breast cancer at IRUA levels is observed by Global Moran's Index (I 0.404/ $p$ 0.01; I 0.555/ $p$ 0.01). From the calculation of LISA, it was possible to identify the IRUA in function of its statistical significance levels. Fig 2 presents the spatial correlation analyses of the proportion of advanced stage diagnosis and adjusted mortality rates for breast cancer in the Brazilian territory.

Fig 3 shows the spatial correlations observed between the proportions of late stage diagnosis of breast cancer and socioeconomic and health service-related population indicators. Most correlations presented negative values, except for the correlation with Gini's Index. The independent variables "*Density of Mastologists*" (p 0.552) and "*Coverage of Basic Attention*" (p 0.929) did not present a significant correlation according to the correlation matrix.

Fig 4 depicts the spatial correlations observed between the adjusted mortality rates for breast cancer and the socioeconomic and health service-related population indicators. The result of the correlation matrix indicates that all independent variables studied herein presented statistically significant correlation with the adjusted mortality rates for breast cancer, except for "*Density of Mastologists*" (p 0.967) and "*Coverage of Basic Attention*" (p 0.262)

Table 2 presents data of the spatial regression analyses for the proportion of late stage diagnosis and adjusted mortality rates for breast cancer, per IRUA. The final spatial model for the

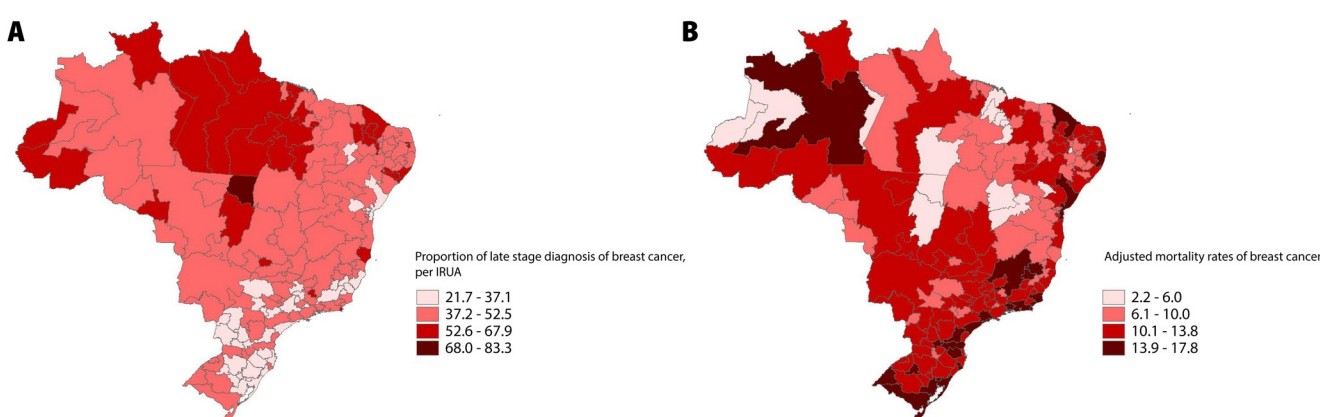

**Fig 1. Spatial distribution of the proportion of late stage diagnosis and adjusted mortality rates for breast cancer in the IRUA, for 2011–2015.**

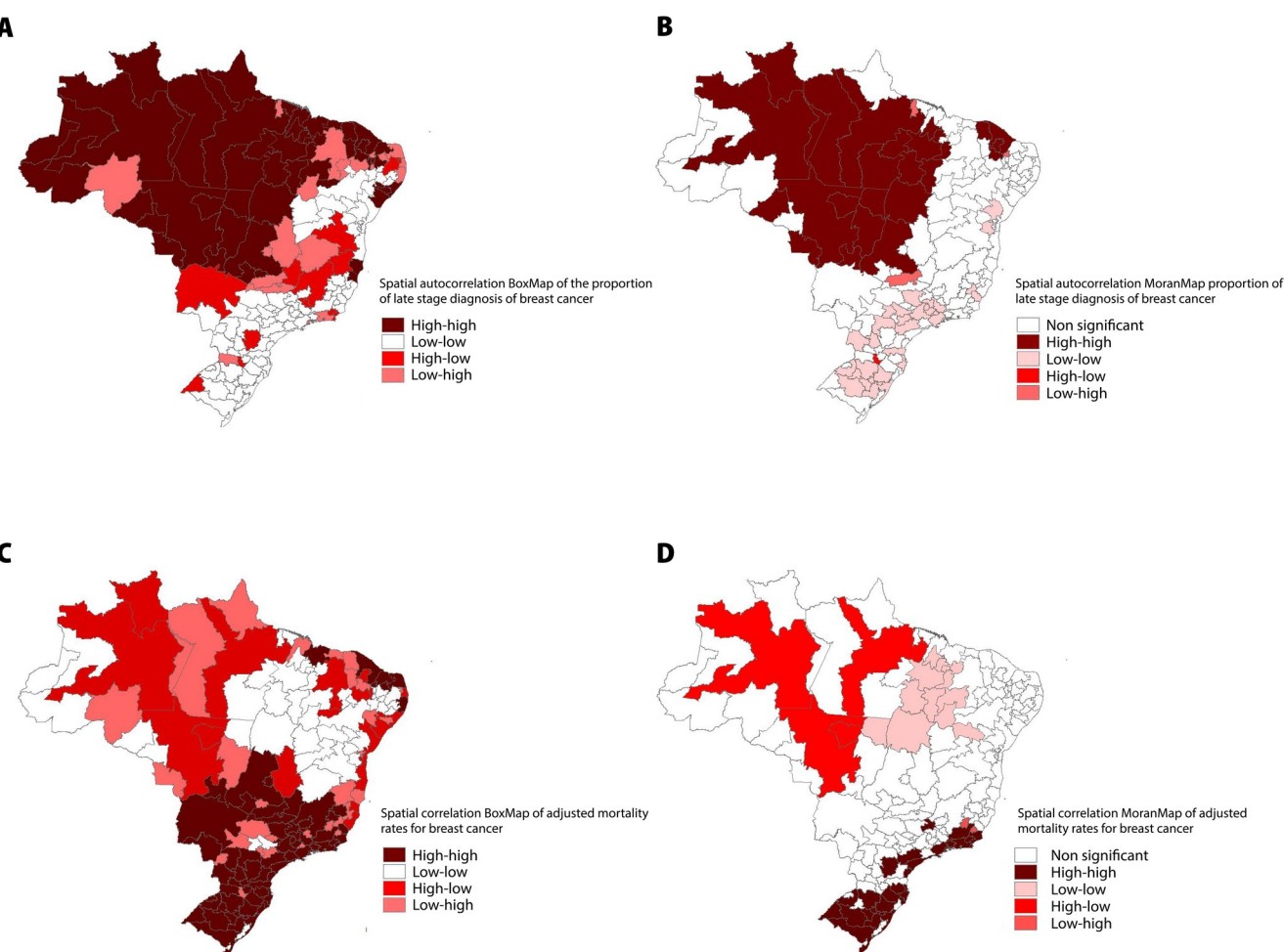

**Fig 2. Spatial distribution of the clusters of proportion of advanced stage diagnosis and adjusted mortality rates for breast cancer with global and local indicators of spatial association, per IRUA, 2011–2015. (A)** BoxMap of the proportion of late stage diagnosis of breast cancer; **(B)** MoranMap of the proportion of late stage diagnosis of breast cancer. Moran's I Moran 0.5549; p 0.001; **(C)** BoxMap of the adjusted mortality rates for breast cancer; **(D)** MoranMap adjusted mortality rates for breast cancer. Moran's I 0.4036; p 0.001.

analysis of the proportion of late stage diagnosis of breast cancer included Gini's Index and the indicators of health service offer "*Density of Gynecologists*" and "*Density of Mammographic equipment*". The model for the analysis of breast cancer mortality was composed of the HDI socioeconomic indicator and indicators related to the offer of health services (*Density of Gynecologists*" and "*Density of Mammographic equipment*"). The variable "*Density of Mammographic equipment*" remained in both models, despite not presenting statistical significance, due to its theoretical plausibility and capacity of statistical fit. Some variables that presented statistical importance in bivariate spatial analysis were not inserted in the model due to the presence of collinearity with other variables already included.

The multivariate model of spatial regression for analysis of late stage diagnosis of breast cancer has an explanatory power of 34.3%. The analysis model for breast cancer mortality presented an explanatory power of 65.7%. These models showed the highest likelihood values and lowest values for Akaike's and Schwarz's Information Criteria. The residues of the models presented normal distribution, and Global Moran's I was -0.025 (p0.344) for the analysis of late stage diagnosis and -0.027 (p 0.333) for the analysis of mortality. S1 Fig and S1 Table, inserted

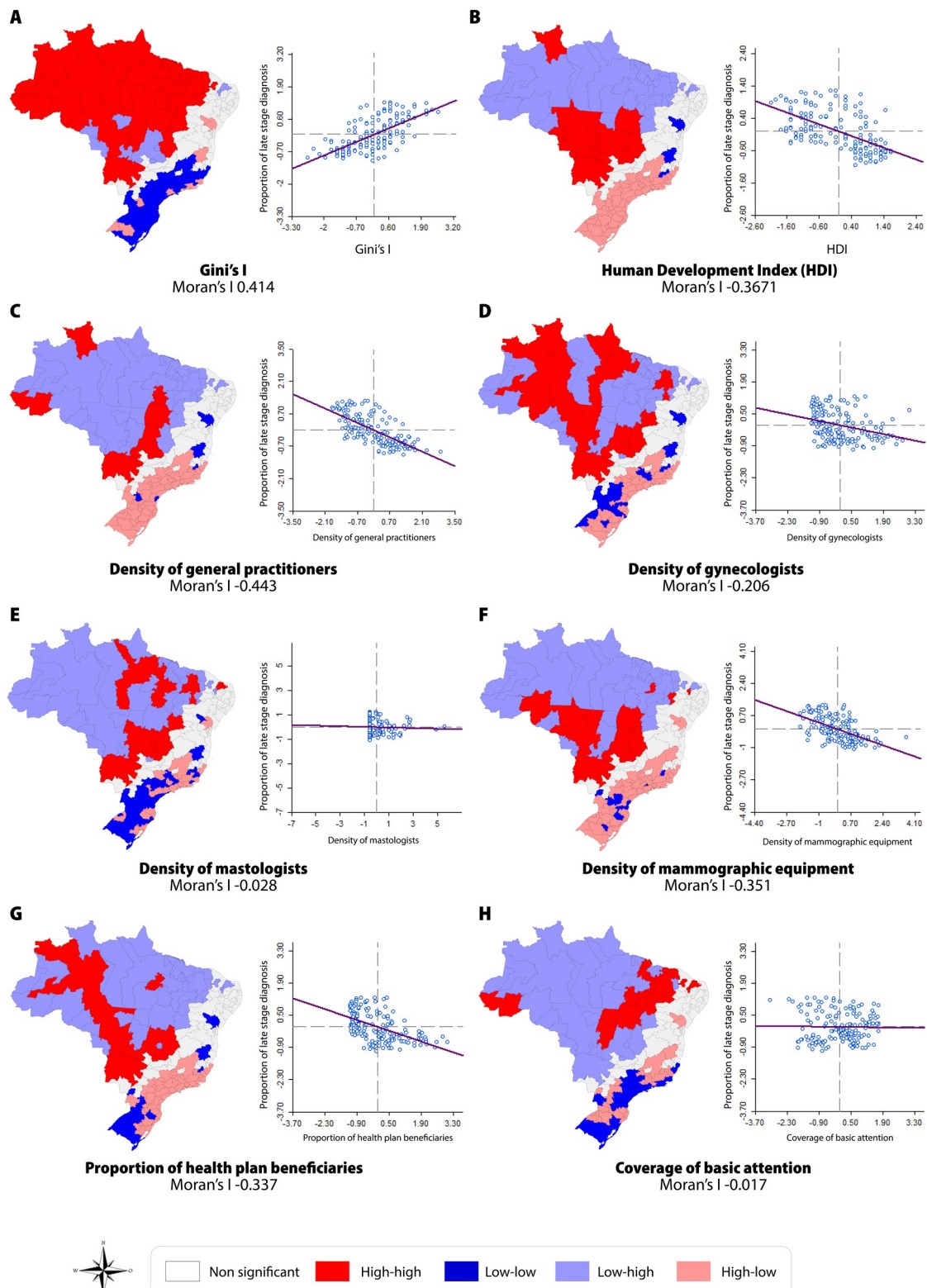

**Fig 3. Spatial correlation between the proportion of late stage diagnosis of breast cancer and socioeconomic and health service-related population indicators, per IRUA, 2011–2015.**

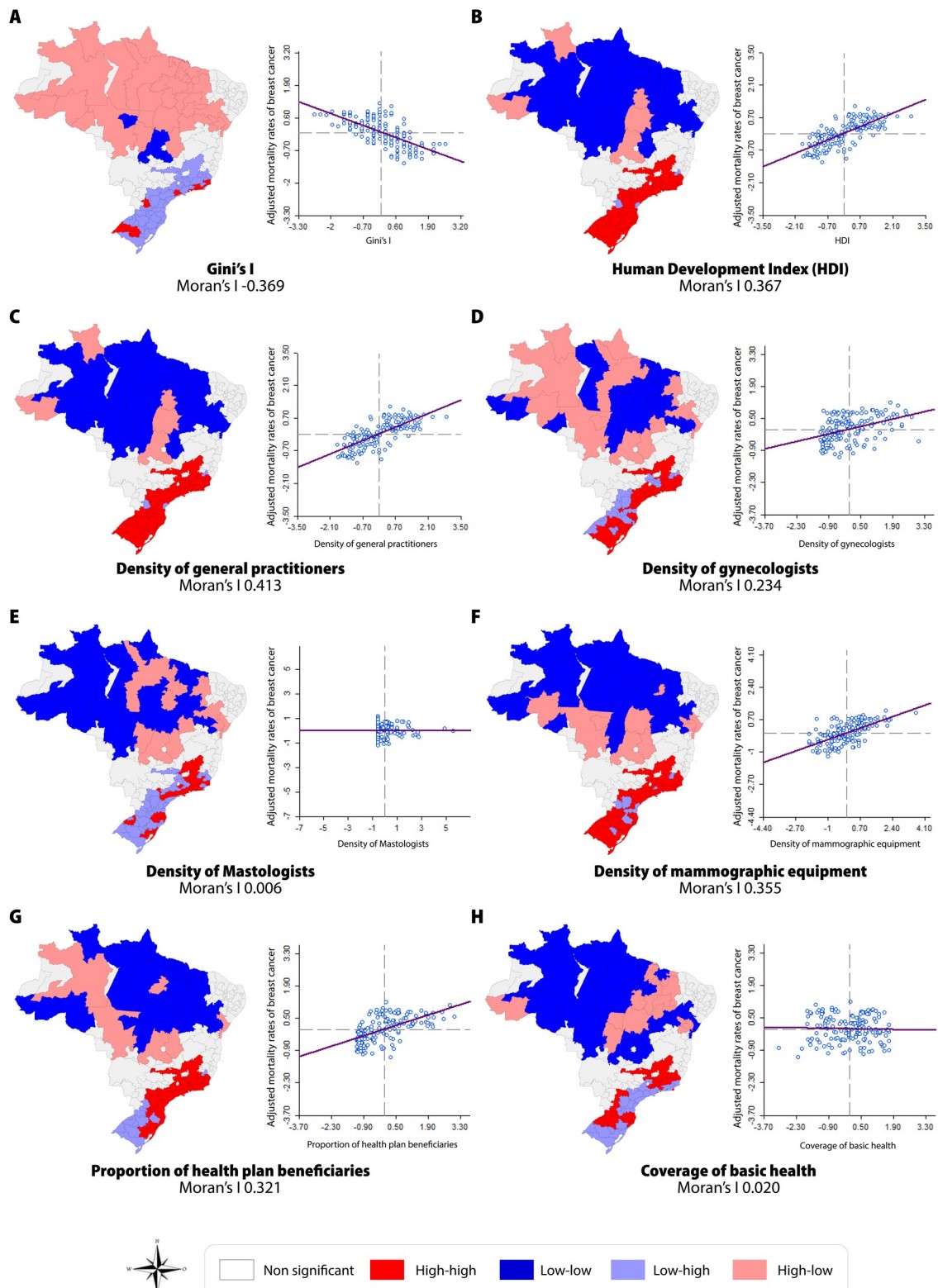

**Fig 4. Spatial correlation between the adjusted mortality rates for breast cancer and socioeconomic and health service-related population indicators, per IRUA, 2011–2015.**

**Table 2. Spatial regression analysis of the proportions of late stage diagnosis of breast cancer and its correlation with socioeconomic and health service offer-related population indicators, per IRUA, 2011–2015.**

| | Coefficient | Standard error | t | p |
|---|---|---|---|---|
| **Late stage diagnosis of breast cancer** | | | | |
| **Socioeconomic population indicators** | | | | |
| Gini's I | 55.11 | 21.47 | 2.57 | 0.010* |
| **Health service-related population indicators** | | | | |
| Density of gynecologists | -0.23 | 0.09 | -2.61 | 0.009* |
| Density of mammographic equipment | 0.02 | 0.43 | 0.05 | 0.956 |
| **Adjusted mortality rates for breast cancer** | | | | |
| **Socioeconomic population indicators** | | | | |
| HDI | 31.82 | 5.78 | 5.51 | <0.001* |
| **Health service-related population indicators** | | | | |
| Density of gynecologists | 0.12 | 0.03 | 4.01 | <0.001* |
| Density of mammographic equipment | -0.05 | 0.12 | -0.44 | 0.660 |

* Statistically significant.

Proportions of late stage diagnosis of breast cancer: Spatial Error Model's R-Squared = 0.453.

Adjusted mortality rates for breast cancer: Spatial Error Model's R-Squared = 0,623.

as supplementary material, present the analysis of residues and compare the values of each regression developed.

Late stage diagnosis of breast cancer presents a positive spatial correlation with Gini's Index (*p* 0.001) and a negative correlation with the density of gynecologists (*p* 0.009). The adjusted mortality rates for breast cancer presented a positive, statistically significant correlation with HDI (*p* <0.001) and with the density of gynecologists (*p* <0.001). In both spatial models, the socioeconomic population indicators presented higher values than the indicators related to health service offer. This indicates the high predictive power of these variables in the statistical models.

## Discussion

The spatial distribution of morbimortality associated with breast cancer presented herein evidences the socioeconomic inequalities across the Brazilian territory. The results demonstrate the presence of spatial clusters in the IRUA located in the North, Northeast, and Midwest Brazil regarding the high proportions of late stage diagnosis of breast cancer. The results suggest that the IRUA with the highest levels of local socioeconomic inequality and lower offer of specialized health services presented high proportions of late stage diagnosis of breast cancer.

The high proportions and unequal territorial distribution of late stage diagnosis of breast cancer verified herein are compatible with previous studies developed in Brazil. The prevalence of late staging for female breast cancer varies between 40.2% and 53.5% and presents regional variations, with the North (48.7%), Northeast (44.5%), and Midwest (47.5%) displaying the highest levels of late stage detection of breast cancer [5,32].

Brazil presents the most extensive public health system in the world, with universal character, aimed at equity and integral care. Approximately 80% of the Brazilian population is assisted exclusively by the national public health system [33]. However, the high demand for healthcare causes the incapacity of the public system to attend the collective health necessities, which leads the population to search for private health services [34]. The unequal territorial distribution of resources and health technology results in the concentration of cancer assistance services in large urban centers of Brazil [35].

The IRUA of the South and Southeast have the best urban organization, with structured health services and orderly distributed across the territory, besides presenting the highest coverage rates of private health plans in Brazil. The North and Northeast regions show irregular population distribution, with large areas presenting low population density, limiting the distribution of health services in the territory. Despite depicting a well-defined territorial occupation, the Northeast concentrates health services and technology in large urban centers the occupy the coastal region, which limits the offer of healthcare and technology to the population of the interior [7,36].

In other countries with different territorial and sociopolitical contexts, it is possible to observe territorial variations associated with late stage diagnosis of breast cancer [37,38]. North-American studies have identified spatial clusters in different states of the USA, with rates of late stage diagnosis of breast cancer varying between 33.5 and 48.2 per 100,000 women. The low socioeconomic conditions and census indicators of poverty have been related to late stage diagnosis of breast cancer in these regions [37,38]. In Iran, areas of territorial clusters have also been studied, with high rates of late stage diagnosis of breast cancer, which presented differences related to access to healthcare and diagnostic delays [39].

The sociopolitical and economic contexts associated with the access to healthcare are considered the main factors contributing to inequalities in morbimortality for breast cancer [7]. In Brazil, the vast extension of the territory and its historical and unequal spatial distribution of municipalities and population have contributed significantly to the contrasts in income distribution in the country [40].

The results of this study indicate a spatial correlation between late stage diagnosis of breast cancer and inequalities related to local income at IRUA levels, measured by Gini's Index. This can be explained by the irregular distribution of financial resources and health services among the municipalities that constitute the IRUA. The scenario leads to the maintenance and increase of inequalities related to the access to health services and, consequently, to the high prevalence of late stage diagnosis of breast cancer in the most unequal areas of Brazil [7,41,42].

Access to health services reflects the inequalities in the distribution of hierarchical levels of assistance to cancer patients [43,44]. The density of gynecologist doctors presented herein acts as a proxy to analyze the general access of the female population to services related to women's health.

Data indicate that the low offer of gynecologists is associated with higher rates of late stage detection of female breast cancer [45]. Findings of a Brazilian study have revealed that the access to gynecological assistance in the last two years and regular Papanicolaou tests lead to higher levels of information and early detection of breast cancer in Brazilian women [46]. The association between early detection and access to gynecologists can be explained by a higher adherence to breast cancer screening programs [47].

This study showed territorial clusters with high adjusted mortality rates located in the IRUA of the South and Southeast. These regions present high levels of global socioeconomic development and a wider offer of intermediate-level healthcare. The South and Southeast regions present the highest incidence rates of breast cancer in Brazil, with an estimated risk of 81.06 and 71.16 per 100,000, respectively [4].

In low- and intermediate- income countries, it is possible to observe a change in the profile of breast cancer morbimortality, especially in the displacement of diagnosis related to poverty and cancer-related infections to areas with higher development. This observation is associated with the processes of population increase and aging, accompanied by alterations in the distribution and prevalence of cancer risk factors [1].

A previous study developed in Brazil, in the South, shows a positive spatial correlation between high mortality rates for breast cancer and better socioeconomic conditions and access

to healthcare in the municipalities [7]. The correlation described between mortality and high levels of global development can be related to the reverse causality idea. In more developed regions, with a better offer of health services and technology, the number of breast cancer diagnoses is higher. Consequently, there is a higher mortality burden for the disease [48].

The results of this study are discussed based on the Law of Inverse Care. This law is the result of policies that limit the access of the population to healthcare in such a manner that the availability of health services is inversely proportional to the necessities of the population [49,50]. The most vulnerable women, living in more developed areas with a higher concentration of population, face difficulties in obtaining health assistance related to prevention, diagnosis, and treatment of breast cancer [7]. This fact suggests s direct relationship with high mortality rates for breast cancer in these regions.

The scenario constituted by territorial regions with higher development levels and better availability of resources and the high costs associated with modern cancer treatment options can restrict the offer and access to these technologies [51]. The inaccessibility to modern treatment options for breast cancer, which are effective but more expensive, affects the health outcomes related to the disease.

The spatial correlation between breast cancer mortality and the density of gynecologists indicated that the IRUA with higher mortality levels detain or are close to specialized women's healthcare centers, which enhances the secondary prevention strategies for breast cancer.

International studies show an association between the high density of medical professionals and high mortality rates for cancer in countries with low- and intermediate- incomes [51,52]. However, the studies that assess the density of gynecologists in the context of breast cancer are directed to the secondary prevention of the disease, aiming to discuss early detection by mammographic screenings.

The study of Rocha-Brischialiri et al. shows a positive spatial correlation between breast cancer mortality and access to chemotherapy and radiotherapy in Brazil [7]. The recent study by Oliveira et al., who evaluated breast cancer mortality in Brazilian IRUA, evidences that the areas with a higher offer of specialized cancer services and higher density of general practitioners presented high adjusted mortality rates for this neoplasm [48].

Cancer-related studies that focus on its spatial location enable the comprehension of the causal relationships regarding contextual socioeconomic conditions and health-related opportunities of the population, aimed at the offer and access to health services and technologies. The specific analysis of IRUA evaluated the Brazilian territory from an organization that considers the influence regions of cities, establishing territorial flows of access to essential activities and health services in the municipalities. The study presented herein bridges the gap regarding the spatial context of late-stage breast cancer diagnosis and mortality, considering a geographic unit of the Brazilian territory that is scientifically underexplored.

The utilization of secondary sources from health information systems in Brazil can be a possible fragility of this study. The socioeconomic contextual indicators (2010 reference) are a limitation of the Atlas of Human Development in Brazil. However, considering the period analyzed herein, there were no significant changes in the national socioeconomic context. Regarding SIM, there has been a significant improvement in the completeness of epidemiological variables in recent years [53]. Regarding cancer staging, IRHC is the most complete secondary source of data in Brazil, reuniting epidemiological data of the main hospital units providing cancer assistance in the country. It is relevant to ponder the possibilities of spatial analyses considering other Brazilian territorial organization units and other determinant factors regarding the health of individuals and their collectivities. There are more disaggregated geographic units in Brazil–however, these are challenged by issues related to coverage and data registry, which at the end compromise the quality of the information generated.

## Conclusions

Based on the geographic information presented, the socioeconomic and health service-related inequalities in the Brazilian territory are determinants of the spatial pattern of morbimortality for breast cancer in the country. The areas with higher needs and worst health assistance conditions are marked by high indices of late stage diagnosis of breast cancer. The more developed regions, which concentrate services and technology, present high mortality rates due to malignant breast neoplasms.

This study contributes to the establishment and reorientation of public policies aimed at controlling breast cancer in the most diverse realities of the Brazilian territory. Implementing effective tracking programs, timely access to appropriate cancer diagnosis and treatment, and the guarantee of equity and integrality in healthcare can help reach better results regarding morbimortality due to breast cancer in Brazil.

## Supporting information

**S1 Fig. Analysis of residues for the Spatial Error Model.**
(TIF)

**S1 Table. Comparative data between the spatial regressions of the proportion of late stage diagnosis and adjusted mortality rates for breast cancer.**
(DOCX)

## Author Contributions

**Conceptualization:** Nayara Priscila Dantas de Oliveira, Dyego Leandro Bezerra de Souza.

**Data curation:** Nayara Priscila Dantas de Oliveira, Luís Felipe Leite Martins.

**Formal analysis:** Nayara Priscila Dantas de Oliveira, Marianna de Camargo Cancela, Luís Felipe Leite Martins, Dyego Leandro Bezerra de Souza.

**Investigation:** Nayara Priscila Dantas de Oliveira.

**Methodology:** Nayara Priscila Dantas de Oliveira, Marianna de Camargo Cancela, Luís Felipe Leite Martins, Dyego Leandro Bezerra de Souza.

**Project administration:** Dyego Leandro Bezerra de Souza.

**Resources:** Nayara Priscila Dantas de Oliveira.

**Writing – original draft:** Nayara Priscila Dantas de Oliveira, Luís Felipe Leite Martins.

**Writing – review & editing:** Nayara Priscila Dantas de Oliveira, Marianna de Camargo Cancela, Luís Felipe Leite Martins, Dyego Leandro Bezerra de Souza.

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
