## [Decision Letter · Decision Letter 0]

17 Dec 2020

PONE-D-20-34094

SPATIAL DISTRIBUTION OF ADVANCED STAGE DIAGNOSIS AND MORTALITY OF BREAST CANCER: SOCIOECONOMIC AND HEALTH SERVICE OFFER INEQUALITIES IN BRAZIL

PLOS ONE

Dear Dr. de Souza,

Thank you for submitting your manuscript to PLOS ONE. After careful consideration, we feel that it has merit but does not fully meet PLOS ONE’s publication criteria as it currently stands. Therefore, we invite you to submit a revised version of the manuscript that addresses the points raised during the review process.

This is a very interesting and important paper. The paper needs to review some parts to improve the discussion and clarify some points, There are interesting results across regions of the countries that should have a more detailed discussion. I also would like to see a more detailed analysis of data quality (some additional references on the quality of information) and limitations of data use (death registration, population estimates). Please, see detailed comments by the reviewers are presented below. 

We look forward to receiving your revised manuscript.

Kind regards,

Bernardo Lanza Queiroz, Ph.D

Academic Editor

PLOS ONE

Journal Requirements:

'This study was financed in part by the Coordenação de Aperfeiçoamento de Pessoal de Nível Superior - Brasil (CAPES) - Finance Code 001.  The funders had no role in study design, data collection and analysis, decision to publish, or preparation of the manuscript'

a. Please provide an amended statement that declares *all* the funding or sources of support (whether external or internal to your organization) received during this study, as detailed online in our guide for authors at http://journals.plos.org/plosone/s/submit-now

Please also include the statement “There was no additional external funding received for this study.” in your updated Funding Statement.

3. We note that Figures 1, 2, 3, 4 and S5 in your submission contain map images which may be copyrighted.

a. You may seek permission from the original copyright holder of Figures 1, 2, 3, 4 and S5 to publish the content specifically under the CC BY 4.0 license. 

Reviewers' comments:

Reviewer's Responses to Questions

**Comments to the Author**

1. Is the manuscript technically sound, and do the data support the conclusions?

Reviewer #1: Yes

Reviewer #2: Yes

2. Has the statistical analysis been performed appropriately and rigorously? 

Reviewer #1: Yes

Reviewer #2: Yes

3. Have the authors made all data underlying the findings in their manuscript fully available?

Reviewer #1: Yes

Reviewer #2: Yes

4. Is the manuscript presented in an intelligible fashion and written in standard English?

Reviewer #1: Yes

Reviewer #2: Yes

5. Review Comments to the Author

Reviewer #1: Dear Editor:

Thank you for offering me the opportunity to contribute to the review of this study that may allow new perspectives to lessen the severe statistics of breast cancer.

This article envisions the spatial distribution pattern of the final stage of breast cancer diagnosis and mortality and its correlation with socio-economic aspects and population indicators related to the provision of health services. The theme is relevant, bring in robust data and the results found may reflect the high incidence of breast cancer mortality of the above-mentioned country. I also highlight that some variables assessed contributes greatly to reflect on the improvement of public health policies concerning breast cancer. Another highlight is that spatial analyzes in continental countries are essential for improving health policies due to the faced social barriers. Thus, I make some suggestions to the authors that may contribute to the disclosure of the study.

Introduction

In general, the introduction is relevant to data that refer to the objective of the study. I only suggest adding data from first world countries in the introduction show emphatically the magnitude of breast cancer in the country under study. Also, I suggest including data on breast cancer survivors diagnosed in late breast cancer diagnoses. These data would add even more relevance to the study.

Methods

1- I would suggest replacing “Materials and Methods” with “Methods”.

2- The section presents the distinct data that helps in understanding the methodological path. However, it is still unclear how the IRUA representation in data collection happened for a continental country like Brazil. I suggest adding a concise statement of this extent collection and the importance of using it in the study. And I would like to highlight positively that the authors have made the rate adjustments.

3- Also, it would be suitable for the authors to explain the socioeconomic and demographic indicators selected to be analyzed in the research.

4- The authors searched for mortality data up to 99 years of age. The suggestion is for the authors to explain the reason for not using a lower age range of mortality for the study and instead of seeming like a limitation of bias due to elderly women having other impairments, it may highlight a gap for older women’s lack of care concerning the disease.

Discussion:

This is a consistent research and can adequately explain the results found.

Greater concerns

1. In general, there was no explanation of the negative results found for the South and Southeast. For greater robustness of the study, the authors could give this information at the beginning of the discussion just as they did with the other regions of the country. There is no relation between socioeconomic variables and lack of professionals according to the other regions, but they show relevant data for these two regions that stand out the most in breast cancer mortality. Subsequently, a deeper interpretation of these findings is necessary.

Minor concerns:

1. Page 15, line 334-337: Although a limitation of the study was described in the paragraph, there is no link from the negative to the positive highlight of the information system. Furthermore, I suggest reviewing the writing so there is a continuity of data about the information system so that the content is globally understandable.

3. Page 15, line 337-339: there was an emphasis on other spatial analysis arrangements and other health determinants. As a suggestion to the authors, there is a need to be more specific in the gap signaled. Thus, researchers will be more supported to produce new research when reading this study.

Reviewer #2: Title: SPATIAL DISTRIBUTION OF ADVANCED STAGE DIAGNOSIS AND MORTALITY OF BREAST CANCER: SOCIOECONOMIC AND HEALTH SERVICE OFFER INEQUALITIES IN BRAZIL

First, the reviewer declares that no competing interests exist.

The article aims to analyze the spatial distribution pattern of late stage diagnosis and mortality for breast cancer and its correlation with socioeconomic and health service offer-related population indicators. Mortality data were collected from the Mortality Information System (MIS). Tumor staging data were extracted from the Hospital Cancer Registry (HCR). Socioeconomic variables were obtained from the Atlas of Human Development in Brazil. Global Moran's Index and Local Indicator of Spatial Association (LISA) were utilized to verify the existence of territorial clusters. Multivariate analysis used models with global spatial effects. The author found a very important result for the implementation of health policies to combat breast cancer: the proportion of late stage diagnosis of breast cancer was 39.7% (IC 39.4 – 40.0), and the proportion of late stage diagnosis presented positive spatial correlation with Gini’s Index (p = 0.001) and negative with the density of gynecologist doctors (p = 0.009).

The authors begin the article presenting information about the incidence of breast cancer in the world, and the authors contextualize the Brazilian case very well, when they argue that health-related inequalities related to the diagnosis and mortality of breast cancer are affected by contextual socioeconomic conditions and the offer and access to health services, in turn related to high social and income-related inequalities. Therefore, they carry out an updated bibliographic review.

In the section “Materials and Methods”, the first subsection “Study design and participants” could just be called “Study design”, because the authors presented only the selected unit of analysis. The choice for Immediate Geographic Regions of Urban Articulation (IGRUA) is justified due to its representativeness of the urban articulations and contextual reality of the period. In Brazil, there are more disaggregated geographical units, such as the micro-regions or even the municipalities. It is important that the authors justify the choice of the geographical unit in view of the quality of the data. For example, it was decided not to work with the municipalities as there are problems about quality in working with more disaggregated data, such as registration or coverage problems. It is common the use of statistical methods (such as empirical bayes) for the correction of records, in small areas. Therefore, even if such problems have not been identified, it is important to contextualize a little more the choice by IGRUA, in view of the other geographical units.

In the section “Study variables and data sources” a table must be created with the variables used, data sources and year of information. In this section, the data collected from ICD 10 were presented, obtained from the Brazilian Mortality Information System (MIS). The methods used to correct the data are adequate. I just suggest the citation of Preston et al (2001): Demography, Measuring and Modeling Population Processes, for direct standardization. In relation to the population estimates used for the denominator of rates, the IBGE estimates were used. There is no necessity of change, no doubt it is a good option, but best population estimates in Brazil today are from UFRN, from the Department of Demography and Actuarial Sciences. This group has been a reference in Brazil (including in IBGE committees) in the development of stochastic municipal projections, by sex and age. There is a project, called “Brazil three times”, financed by the Secretariat of Strategic Affairs of the Presidency of the Republic, which provides estimates (by sex and age) that are more robust, when compared to IBGE estimates. You do not need to change this information in the article, but this is a good option for the next publications.

In addition, in relation to the data processing stages, the authors are very careful, explaining how they dealt with situations of lack of data, missings, etc. I only suggest that a highlight is given to the fact that data on inequality were obtained from the Atlas of Human Development in Brazil for 2010, while mortality rates, for example, were estimated for the period between 2011-2015. Of course, it is a good assumption to consider that the socioeconomic context did not change significantly, for example, between 2010 and 2015. But these assumptions (which represent limitations of the data sources) need to be described in the methodology.

Regarding the geographic information system used, Terraview, it is a good GIS, created by INPE. But the maps need to be corrected. It's necessary to export them with a higher resolution (.dpi), since the low resolution is compromising the visualization, mainly of the thematic maps. This is a requirement for the publication of this article.

Regarding the cluster maps, it was evident that the authors printed the maps and graphs from the Geoda. There are ways to export the table of attributes of the results of spatial autocorrelation, to the elaboration of a more complete layout, in a more robust GIS. But, in this case, you don't need to change the cluster maps for the publication.

In the subsection “Statistical analysis”, both the auto correlation method and the spatial regression method were not detailed, but references were cited that allow the reader to known the methods, if interested. On the other hand, the parameters and procedures adopted in the models were presented (such as the neighborhood matrix and the residue analysis). Therefore, this section is well structured and does not need correction.

The “Results” section is well structured, the results are very interesting, and represent an important contribution to this field of knowledge. In Brazil, 39.7% of cases are diagnosed late, and statistical significance was found between late care and the incidence of the disease with inequality and the presence of specialized professionals. The “Discussion” section, on the other hand, analyzes the spatial inequalities observed in different regions of the country, putting in perspective the inequality in the offer of services (concentrated in large centers), added to the problem of deficiency in the public health service, responsible for attending 80% of the population. These two sections analyzed the results satisfactorily, in addition to highlighting the importance of the study, so there is no indication of correction in these sections.

The article is of great quality and contributes to the studies on breast cancer. I strongly suggest publication, after minor corrections.

6. PLOS authors have the option to publish the peer review history of their article (what does this mean?). If published, this will include your full peer review and any attached files.

Reviewer #1: **Yes: **SHEILA CRISTINA ROCHA BRISCHILIARI

Reviewer #2: **Yes: **Járvis Campos - Professor of Demography and Acturial Sciences (UFRN) and the Demography Graduate Program (PPGDem/UFRN)

---

## [Author Response · Author response to Decision Letter 0]

13 Jan 2021

All the suggestions made by the editor and reviewers were followed, aiming to further increase the quality of the paper.

The images were exported at higher resolution (600 dpi) for individually uploaded images. When converted to the PDF format, quality is lost. The resolutions of all images have been verified and edited to enhance the visualization of maps.

The maps displayed in the manuscript were elaborated by the authors, from territorial geographic mesh (shape files), publicly available from the Brazilian Institute of Geography and Biostatistics (IBGE), and therefore free from copyright. The geographic meshes are made available without any epidemiological associated data. The maps related to late-stage diagnosis and breast cancer mortality per Brazilian IRUA were also elaborated by the authors.

Website of the territorial meshes https://www.ibge.gov.br/geociencias/organizacao-do-territorio/malhas-territoriais.html

REVIEWER #1

Introduction

The requested information has been added, with emphasis on incidence data for breast cancer in first-world countries and survival rates, adding relevance to the study.

Methods

1- The title of the section was changed, following your suggestion. 

2- The representation of IRUA for data collection has been better explained in the paper. The IRUA correspond to an intermediate territorial scale between Federation Units (FU) and Immediate Geographic Regions of Urban Articulation. Following the federation principles, each FU must contain at least two IRUA. This territorial design describes the regions of influence of the main urban centers. In the study, all variables were collected per municipality and then grouped per IRUA. The socioeconomic population indicators were collected per municipality and then grouped per IRUA, using the weighted average of the population.

3- All indicators analyzed in the study were presented in Table 1, as suggested. 

4- The mortality data presented in the study were gathered based on the place of residence and detailed age group, for the period 2011-2015. The malignant breast neoplasm cases were collected for the age rage 18-99 years old, also following the detailed age group logic. The addition of elderly women aimed to verify an association between late-stage diagnosis and mortality with oncology assistance provided by the Brazilian network of healthcare. The study aims to evidence gaps in the offer of healthcare to older women, regarding breast cancer. It was possible to observe that 4.21% of the studied population belonged to the age group 80-89 years old and 0.46% to the group 90-99 year old .

Discussion

1- The negative results related to the territorial clusters of the high mortality rates located for the South and Southeast Brazilian IRUA are presented and discussed starting at page 15,line 313. The discussion was divided into two parts: an initial section related to the findings associated with late-stage diagnosis of breast cancer, and the second section focuses on the discussions of mortality results. The negative results found for the South and Southeast regions of the country are discussed based on the Reverse Causality Theory and Inverse Care Law.

Minor concerns:

1 -3- The information suggested has been added to the new, revised version of the manuscript, considering the potentialities and fragilities of the study. 

REVIEWER #2

Methods

1- The first subsection of “Methods” was changed to “Study Design”, as suggested by the reviewer. Justification of the use of IRUA with emphasis on data quality is presented in the methodology and then in the discussion of results, evidencing the potentialities of the study. 

2- A Table containing the dependent and independent variables of the study was added to the paper (Table 1). 

The reference “Preston et al (2001): Demography, Measuring and Modeling Population Processes” was added (Ref 19).

Regarding the population estimations, IBGE data was utilized as it is a widely accepted and employed database, being the primary data source. The suggestion of the reviewer to use the estimations of the Department of Demographics and Actuarial Sciences has been noted for futures studies.

3- The limitations regarding the data sources of the study are presented in the discussion section (Page 17, Line 359).

4- The images were exported at higher resolution (600 dpi) for individually uploaded images. When converted to the PDF format, quality is lost. The resolutions of all images have been verified and edited to enhance the visualization of maps.

5- We thank the reviewer for the information, and the cluster maps have not been changed. 

6- We appreciate your comments. 

7- Thank you for the time dedicated to the review of our paper.

---

## [Editor Report · Decision Letter 1]

19 Jan 2021

SPATIAL DISTRIBUTION OF ADVANCED STAGE DIAGNOSIS AND MORTALITY OF BREAST CANCER: SOCIOECONOMIC AND HEALTH SERVICE OFFER INEQUALITIES IN BRAZIL

PONE-D-20-34094R1

Dear Dr. de Souza,

We’re pleased to inform you that your manuscript has been judged scientifically suitable for publication and will be formally accepted for publication once it meets all outstanding technical requirements.

Kind regards,

Bernardo Lanza Queiroz, Ph.D

Academic Editor

PLOS ONE

Additional Editor Comments (optional):

Thank you for considering and incorporating the comments and suggestions made during the review process. We all agree the paper is a relevant and important contribution. 
---

## [Editor Report · Acceptance letter]

22 Jan 2021

PONE-D-20-34094R1 

SPATIAL DISTRIBUTION OF ADVANCED STAGE DIAGNOSIS AND MORTALITY OF BREAST CANCER: SOCIOECONOMIC AND HEALTH SERVICE OFFER INEQUALITIES IN BRAZIL 

Dear Dr. de Souza:

I'm pleased to inform you that your manuscript has been deemed suitable for publication in PLOS ONE. Congratulations! Your manuscript is now with our production department. 

Kind regards, 

on behalf of

Dr. Bernardo Lanza Queiroz 

Academic Editor

PLOS ONE